# A Review on Cyanide Gas Elimination Methods and Materials

**DOI:** 10.3390/molecules27207125

**Published:** 2022-10-21

**Authors:** Xuanlin Yang, Liang Lan, Ziwang Zhao, Shuyuan Zhou, Kai Kang, Hua Song, Shupei Bai

**Affiliations:** State Key Laboratory of NBC Protection for Civilian, Beijing, 100191, China

**Keywords:** cyanide, adsorption, catalysis, porous material, metal oxide material, carbon material

## Abstract

Cyanide gas is highly toxic and volatile and is among the most typical toxic and harmful pollutants to human health and the environment found in industrial waste gas. In the military context, cyanide gas has been used as a systemic toxic agent. In this paper, we review cyanide gas elimination methods, focusing on adsorption and catalysis approaches. The research progress on materials capable of affecting cyanide gas adsorption and catalytic degradation is discussed in depth, and the advantages and disadvantages of various materials are summarized. Finally, suggestions are provided for future research directions with respect to cyanide gas elimination materials.

## 1. Introduction

Cyanide gas refers to gaseous substances containing cyano groups, such as hydrogen cyanide (HCN), cyanogen chloride (CNCl) and dicyan ((CN)_2_). The amount of the highly volatile and toxic cyanide gas present in the atmosphere is relatively small, and most of it is produced as a result of human activities and industrial production. Its main sources include the processing and use of cyanide-containing chemicals [1], the combustion of coal-based fossil fuels [1], the pyrolysis of biomass [2], the combustion of building decoration materials [3], and the denitrification of exhaust gas [2,3]. As a precursor utilized in chemical reactions, cyanide gas is widely available and inexpensive, so its use in future wars and terrorist operations cannot be ruled out. HCN and CNCl, which are commonly present in cyanide gas, have been used as systemic toxic agents in the military context. After being inhaled or ingested, the ionized cyanide radicals from cyanide gas combine with the ferric iron of cytochrome oxidase in mitochondria, preventing the iron (III) center in the enzyme from undergoing physiological reduction, thus hindering the normal respiration of cells and leading to tissue hypoxia and asphyxia of the body [4].

In gas masks, protective engineering, and mobile platforms, the elimination of cyanide gas is mainly achieved by forcing the contaminated gas to pass through a fixed bed reactor filled with an adsorbent. Given the small size of cyanide molecules and the high vapor pressure, depending on the physical adsorption of cyanide gas, it is difficult to realize effective removal of cyanogen gases using porous substances such as activated carbon due to the weak interaction between cyanide gas and the adsorbent surface [5]. Thus, adsorbents are usually impregnated with metal salts, which could react chemically with cyanide gas [6]. Nevertheless, the filtering and absorbing elements utilized in existing gas masks, protective engineering, and motorized platforms only ensure short-lasting protection. Cyanide gas emitted into the air as a result of industrial activities can also participate in the formation of nitrogen oxides and acid rain, causing photochemical pollution, destroying the pH balance of the environment [7], and exerting negative impacts on the environment and human health [8]. Therefore, research on the efficient elimination of cyanide gas is of considerable scientific significance in the fields of national defense, industrial production, and environmental protection.

Commonly employed methods for cyanide gas elimination include absorption [9], adsorption [10], and combustion [11,12]. In this paper, we review currently implemented cyanide gas elimination methods. Research progress with respect to the development of cyanide gas elimination materials is discussed in detail, and suggestions for future research work in this field are proposed. The scope of this review is shown in Figure 1.

## 2. Cyanide Gas Elimination Methods

### 2.1. Absorption Method

Absorption refers to the process of treating a gas mixture with a liquid absorbent to eliminate one or more gases. The principle of the approach is to eliminate some components of the gas mixture based on differences in solubility or reactivity between the various components with the absorbent [13]. Cyanide gas components, such as HCN and CNCl, exhibit high solubility in water and are acidic. The method of lye absorption refers to the introduction of cyanide gas into an alkali solution (such as sodium hydroxide solution), resulting in a reaction between alkali and cyanide gas. This method can achieve a high cyanide gas elimination rate. This absorption method can be suitably applied to the purification of high concentrations of cyanide gas, and it is characterized by a high processing capacity and high efficiency [14]. However, the absorption method only results in the transfer of cyanide anions into the liquid phase, and it does not achieve the ultimate goal of rendering them harmless. Cyanide ions dissolved in the aqueous phase remain highly toxic. In order to render the absorbed ions completely harmless, post-treatment methods need to be implemented, such as biodegradation, membrane separation, electrolytic oxidation, alkaline chlorination, pressurized hydrolysis, or oxidation by a strong oxidant [15]. These post-treatment methods require energy and chemical reagents, which are costly. The treatment of excessive chemical reagents in the absorption solution is also a serious challenge in terms of environmental protection. The absorption liquid of cyanogen gas is strongly alkaline and corrosive. It places high demands on the anticorrosion performance of the equipment. Therefore, equipment requires additional maintenance, representing another expense. Absorption methods also involve long processing cycles, often limiting their practical application [16]. Therefore, more efficient and environmentally friendly methods for cyanide gas elimination need to be developed in the future.

### 2.2. Combustion Method

Cyanide gas is usually accompanied by combustible components, such as CO, H_2_, and hydrocarbons. Through combustion, cyanide gas and the other mentioned gases can be completely converted into harmless CO_2_, N_2_, and H_2_O. This method has two advantages. First, toxic and harmful gases can be completely converted into non-toxic substances. Furthermore, the heat generated by combustion can be reused. The approaches implemented in the combustion method can be divided into two categories: direct combustion and catalytic combustion. When the exhaust gas contains a small amount of oxygen, direct combustion can be achieved, as the gas mixture can be simply ignited to convert the cyanide exhaust gas into nitrogen, carbon dioxide, and water. When implementing the direct combustion method, the concentration of the gas during the combustion reaction needs to be controlled, and the change in the composition of the reaction gas needs to be monitored in order to avoid the risk of explosions. In addition, the implementation of the direct combustion method requires high temperatures. The direct combustion process is usually conducted at 80~900 °C. Increasing the temperature is beneficial for the complete conversion of cyanide gas [17]. The direct combustion method can completely convert cyanogen gas without selectivity. The exhaust gas from combustion is generally not environmentally friendly. Additionally, it may cause a waste of energy due to the high combustion temperature. In catalytic combustion, oxygen in the air is activated with the help of the catalyst, reducing the activation energy of the oxidation of cyanide gas is and the eliminating cyanide gas at a lower temperature than in the direct combustion process. However, the high-efficiency catalysts utilized in the catalytic combustion of cyanide gas are mostly expensive noble metals, limiting the application of this method. Although low-cost catalysts, such as CaO, as well as a series of metal-modified zeolites, including Cu–(Beta, FER, MCM–22, MCM–49, MOR) and M–(M=Cu, Co, Fe, Mn, Ni)–ZSM–5, can also be effective for the catalytic combustion of cyanide gas, the relevant catalysts lack environmental stability, especially in high concentrations of water vapor, carbon dioxide, and oxygen [18,19]. However, in practical application scenarios, high concentrations of water vapor, oxygen, and carbon dioxide are often present, reducing the practicability of the catalytic combustion method. In addition, sulfur-containing compounds and other organic gases can be present in industrial waste gas; these species can easily poison the catalyst [20]. Maintaining catalyst activity and reducing the extent of catalyst poisoning during the reaction process are important goals to achieve in the development of catalytic combustion catalysts.

### 2.3. Adsorption Method

The adsorption method can be realized at room temperature with simple and convenient operation and is commonly utilized in the treatment of industrial waste gas, owing to its practicality. The applied approaches can be divided into two categories based on the type of adsorption: physical adsorption and chemical adsorption. At its core, the adsorption method is based on the selection of a suitable high-efficiency adsorbent to achieve elimination of cyanide gas by physical or chemical adsorption. Commonly used adsorbents include activated carbon, zeolites, and molecular sieves, which are easy and relatively inexpensive to prepare. Notably, the performance of these adsorbents can be significantly improved by loading active components onto them [21,22]. The adsorption method is operationally easy to implement, does not require large amounts of aqueous alkali, is highly safe and environmentally friendly compared with the absorption method, and can be suitably applied for the treatment of low-concentration cyanide-containing waste gas. However, the adsorption capacity of the adsorbent is limited, so the processing capacity afforded by the adsorption method is limited. Therefore, the adsorbent needs to be replaced or regenerated frequently. In addition, the cyanide gas desorbed from the adsorbent remains toxic and harmful to the human body and the environment. Further postprocessing of the adsorbed gas is therefore required to satisfy legal airborne permissible exposure limits [23,24]. As a consequence, the use of this method for cyanide gas remediation is limited.

### 2.4. Catalytic Method

The adsorption capacity afforded by pure physical adsorption is limited, and any toxic molecules adsorbed on the relevant adsorbent bed are easily desorbed, causing secondary pollution [25]. In the industrial context, the catalytic decomposition method is a widely employed approach for the treatment of gaseous pollutants, and it generally needs to be carried out at high temperatures. This method can achieve complete conversion of the adsorbed toxic and harmful molecules into harmless substances. Pt/Al_2_O_3_ is a typical high-efficiency catalyst that exhibits high catalytic activity for HCN degradation [26,27]. However, it is associated with high energy and catalyst production costs. Additionally, a considerable proportion of N_2_O and NO_x_ is generated in the degradation reaction, causing secondary pollution, making the method environmentally unfriendly.

## 3. Cyanide Gas Elimination Materials

With respect to the adsorption and catalytic methods, the adsorbent or catalyst material is the core of the approach to the treatment of toxic and harmful gases. The adsorption or catalytic performance of the utilized material directly determines its ability to eliminate the target pollutants. Research progress with respect to the development of materials for cyanide gas elimination is discussed from two perspectives: the development of carbon-based and non-carbon-based materials.

### 3.1. Carbon-Based Adsorbent Materials

The carbon-based materials used in the field of toxic and harmful gas elimination include activated carbon, impregnated activated carbon (activated carbon impregnated with various active components), and activated carbon fibers. Activated carbon is currently the most widely used adsorbent. It is characterized by a large specific surface area and comprises an abundance of micropores and mesopores. These micropores can adsorb macromolecules in the vapor phase. On the other hand, mesopores are often used to load various active components to improve of the comprehensive performance of activated carbon.

Impregnated activated carbon is mainly used for the elimination of harmful gases and vapors. The elimination of cyanogen chloride by impregnated activated carbon can be achieved at room temperature. It is widely used in both industry and military fields. Impregnated activated carbon is used in gas masks and collective protective filters in the military. With its rich microporous structure, impregnated carbon exhibits good adsorption performance toward macromolecules and high-boiling-point vapors. However, the capacity of impregnated carbon to physically adsorb small molecules, such as those present in cyanide gas (e.g., HCN and CNCl), is not strong, and the molecules adsorbed on the material’s surface easily undergo desorption at room temperature [28,29]. Although the impregnated active components can eliminate cyanide molecules by catalyzing a cyanide gas degradation reaction, practical problems exist, such as easy penetration of the material bed, short protection time, easy aging, and deterioration under storage conditions [5,30]. Therefore, new materials need to be designed and developed to simultaneously achieve efficient adsorption and rapid degradation of cyanide gas; these materials need to be used in conjunction with existing materials to make up for the deficiencies in the cyanide gas purification capacity of the latter.

The impregnated carbon materials currently utilized in military applications can be divided into two categories according to the elements contained in the active components: chromium-containing and chromium–free impregnated carbon materials. Chromium-containing impregnated carbon materials, which are represented by ASC and ASC–T carbons (ASC and ASC–T are the code names for impregnated activated carbon. There are no public full names.), exhibit good protection performance against cyanide gas, and their main active components are copper, chromium, and silver ions [31]. However, ASC carbon can easily adsorb water vapor and carbon dioxide at room temperature, which leads to a decrease in its activity. This process is called the aging of impregnated carbon. In the case of ASC–T carbon, the adsorbent material’s ability to resist aging is significantly improved as a result of the addition of triethylenediamine. In addition, toxic substances are adsorbed on the surface of the ASC and ASC–T carbon, then undergoing a reaction processes that converts them into stable compounds, causing the adsorbent material to lose its catalytic activity. Alternatively, the side reaction products can end up covering the surface of the catalyst, thereby preventing contact between catalyst and reactant, resulting in the poisoning of the catalyst [29]. For example, in the catalytic hydrolysis of CNCl on the surface of impregnated carbon, the HCl and NH_3_ molecules generated by the reaction can generate ammonium chloride, which can cover the catalyst surface and block the pores of the adsorbent. Moreover, CNCl can react with the impregnating components. Nonvolatile substances, such as copper chloride and chromium trichloride, are generated. This process leads to the destruction of the active sites on the surface of the catalyst and renders the catalyst ineffective [29].

The active-component chromium loaded on ASC–T carbon is harmful to the human body and can cause environmental pollution during the preparation process. ASZM–T carbon (name of a chromium-free carbon) is the most widely used chromium-free impregnated carbon. It contains molybdenum and zinc in place of the carcinogenic chromium utilized in chromium-containing impregnated carbon and ensures the safety of the wearer. However, the process by which the activated carbon is impregnated with macromolecular molybdenum to produce ASZM–T carbon can easily lead to the impregnating material occupying a large number of pores in the activated carbon support, negatively affecting the comprehensive protective performance of the material and limiting the synergy between the active components. Moreover, molybdenum is expensive, and the cost of mass production is high.

Ting Zhao et al. investigated zirconium-loaded impregnated carbon developed to affect the adsorption and degradation of HCN and CNCl present in the gas phase [32]. This impregnated carbon with coconut shell crushed activated carbon as the matrix material was loaded with 15–20% (mass fraction) Cu, 1–6% Cl^−^, 0.1–5% W, 0.1–5% V, 0.1–5% Ce, and 2–8% Zr, with the remaining material consisting of activated carbon. Implementing the method of equal amount impregnation, after the impregnation solution of the metal salts was prepared, it was evenly poured onto the activated carbon carrier, and the obtained mixture was stirred continuously. After impregnation, the mixture was placed in a sealed bag, left standing for 1–4 h at room temperature, and dried and activated in a hot air flow at 100–160 °C for 24 h. The prepared zirconium-loaded impregnated carbon achieved an elimination effect on cyanide gas. To a certain extent, the materials prepared in this work make up for the shortcomings of the existing impregnated activated carbon materials, which cannot purify small cyanide molecules. Compared with other known impregnated carbon materials, the types of active components loaded on the carbon prepared by this method exhibit a crucial difference: they do not contain harmful metal chromium, and they are environmentally friendly and safe for human health. Furthermore, the use of macromolecular molybdenum and strongly acidic components is avoided, and the preparation process is simple and can be easily adapted to achieve mass production; therefore, this method has considerable application prospects.

Carbon-based materials have some shortcomings. Although activated carbon has a large specific surface area, its pores are uncontrollable and have no fixed size. The structure of activated carbon is not easy to design and control, and it is considerably affected by raw materials (the pore structure of activated carbon differs depending on the raw materials used for production, such as coconut shell carbon and coal-based carbon [33]). The performance consistency of the produced impregnated carbon is uncontrollable. Additionally, activated carbon exhibits good physical adsorption performance toward macromolecular substances but weak physical adsorption performance toward small molecules and low-boiling-point substances. It is difficult to achieve broad-spectrum adsorption of various gases. In addition, the adsorbed molecules remain in the impregnated carbon bed, and their desorption can easily cause secondary pollution. Moreover, activated carbon is sensitive to humidity; specifically, when the relative humidity is greater than 80%, the competitive adsorption of water can result in the desorption of the molecules that have been adsorbed on the carbon, and the fixed bed of adsorbent will lose efficacy immediately [34].

### 3.2. Non-Carbon-Based Adsorbent Materials

In recent years, researchers have actively explored non-carbon-based protective materials in view of the problems associated with the use of traditional carbon-based materials in an effort to address the shortcomings of carbon-based materials. Relevant studies have been conducted on materials such as zirconium hydroxide, Pt/Al_2_O_3_ catalysts, mesoporous molecular sieves, microfiber coating materials, magnetic nanoparticles, and metal–organic frameworks. Most of these materials are used powder form, and the production cost is higher than that of impregnated carbon. At present, these materials are mainly in the state of laboratory research and have not been tested for large-scale application. Zirconium hydroxide, mesoporous silicate molecular sieves, metal–organic frameworks, and other new materials under development are expected to be mixed with impregnated activated carbon to form a composite bed, making up for the inability of impregnated carbon to eliminate small molecular gases. The materials that have been tested for CNCl adsorption capacity are summarized according to the existing literature. Due to the differing test conditions and evaluation criteria, we divided them into Table 1 and Table 2.

#### 3.2.1. Zirconium Hydroxide

Zirconium hydroxide is an intermediate in the synthesis of zirconia, with stable properties and a high specific surface area. The surface of zirconium hydroxide comprises an abundance of hydroxyl functional groups, coordination-unsaturated metal cations, oxygen vacancies, and other active centers [38,39,40]. This species displays acid–base amphoteric characteristics. Its surface can be easily functionally modified, and zirconium hydroxide has potential to be utilized for the broad-spectrum elimination of harmful gases [41].

Studies have shown that zirconium hydroxide elimination materials loaded with various active metal components, such as Zn, Co, Ag, and triethylenediamine (TEDA), exhibit an ability to eliminate low-boiling-point toxic substances.

Gregory et al. [36,42,43,44,45] found that the CNCl elimination ability of zirconium hydroxide, which is quantified by measuring the protection time (i.e., the time elapsed from when the airflow enters the packing layer to when the outlet end reaches the minimum harmful concentration to humans) is 1.42 times that determined for ordinary activated carbon. Modified zirconium hydroxide also has good elimination properties toward other toxic industrial gases, such as sulfur dioxide, ammonia, and chlorine gases (Cl_2_, COCl_2_, and HCl). Based on the excellent properties of zirconium hydroxide, Peterson et al. (Edgewood Chemical Biology Center, USA) prepared a TEDA-loaded zirconium hydroxide material [36,46]. Through fixed-bed penetration experiments, they investigated the ability to eliminate low-boiling-point toxic substances, and the mechanism by which TEDA-loaded zirconium hydroxide eliminates CNCl was analyzed by conducting NMR and XPS experiments. Studies indicated that the alkaline conditions on the surface of zirconium hydroxide itself can promote the hydrolysis of CNCl, and the TEDA loaded onto the surface of zirconium hydroxide considerably accelerates the reaction. A series of experiments afforded data on the elimination performance of coconut shell charcoal, coconut shell charcoal impregnated with 10% TEDA, coconut shell charcoal impregnated with 10% TEDA and 10% copper nitrate solution, and zirconium hydroxide impregnated with 6% TEDA. As demonstrated by the data presented in Figure 2, zirconium hydroxide loaded with 6% TEDA was characterized by the longest protection time against CNCl in humid air.

In summary, zirconium hydroxide achieves better performance in terms of CNCl elimination than impregnated activated carbon. In addition, it displays a good elimination ability toward the small-molecule toxic gases that are difficult to eliminate using impregnated carbon; zirconium hydroxide exhibits synergistic effects with surface-loaded active groups, and it has promising potential in the fields of toxic and harmful gas elimination and air purification. However, following the hydrolysis of CNCl, inorganic chlorides and urea byproducts are deposited in and possibly block the pores, adversely affecting the out diffusion of gas molecules. In addition, the industrial production technology of modified zirconium hydroxide is not yet mature, and the elimination of toxic and harmful gases utilizing this material is still in the laboratory research stage, so further studies need to be conducted in the future.

#### 3.2.2. Pt/Al_2_O_3_ Catalyst

Zhao et al. reported that following adsorption on the surface of a Pt/Al_2_O_3_ catalyst, HCN underwent decomposition to produce H^+^_ads_ and CN^-^_ads_ (hydrogen ions and cyanogen ions adsorbed on the surface); notably, after the oxidation reaction, NO_x_ products were generated. When the gas stream contains water vapor, HCN can also generate NH_3_ and CO_2_ through the hydrolysis reaction, and NH_3_ is further oxidized to N_2_ and NO_x_. However, the low selectivity and excessive byproducts associated with this method require subsequent treatment, which is the main reason that the Pt/Al_2_O_3_ catalyst cannot be employed for the elimination of HCN.

Lester and Marinengeli [47] found that the catalytic activities of Pt/Al_2_O_3_ and Pt/TiO_2_ for the degradation of CNCl differed depending on temperature and humidity conditions. At 150 °C, both catalysts achieved the complete degradation of CNCl within 2 h; in contrast, at 75 °C, the catalytic activity of Pt/Al_2_O_3_ was observed to be far inferior to that of Pt/TiO_2_. Agarwal et al. [48] investigated the ability of Pt/Al_2_O_3_ to catalyze the decomposition of CNCl and found that water molecules play an important role in the process of degradation of CNCl molecules. The presence of water molecules reduced the apparent activation energy of the CNCl degradation reaction and improved the conversion rate of the substrate; the presence of water also changed the product selectivity by initiating an affinity substitution reaction of CNCl. The reaction mechanisms in dry (1) and humid (2) conditions, respectively, are as follows: 2CNCl+1.5O_2_→N_2_+Cl_2_+CO+CO_2_(1)
 CNCl+2H_2_O→NH_3_+HCl+CO_2_(2)

CNCl can be catalytically oxidized on the surface of Pt/Al_2_O_3_ at 200 °C–300 °C. The results of the relevant experiments indicate that the conversion rate of CNCl can reach a value of 98% at 375 °C and 170,000 cm^3^·h^−1^·g^−1^ space velocity when the 2.15% Pt/α–Al_2_O_3_ catalyst is charged in a fixed-bed microreactor [27]. The presence of water molecules plays an important role in the catalytic reaction of CNCl degradation; under anhydrous conditions, the catalytic activity of the mentioned catalyst was observed to be very low. In contrast, as shown in Figure 3, the presence of water molecules promoted the conversion of CNCl and led to an improvement in selectivity of the reaction product, favoring the complete oxidation of CNCl to CO_2_ and HCl, with neither CO nor Cl_2_ detected among the reaction products.

Catalytic decomposition technology is suitable for the degradation of almost all chemical agents. Noble metal-based catalysts, such as Pt/Al_2_O_3_, have a long life, and the relevant catalyst beds do not need to be replaced frequently. However, the energy consumption is high, and the cost of the production of noble metal-based catalysts is high.

#### 3.2.3. Mesoporous Silicate Molecular Sieves

Mesoporous silicate molecular sieves are silicate mesoporous materials that are synthesized using cationic surfactants as templates under alkaline conditions. These materials have a regular pore structure. Mesoporous molecular sieves are good catalyst carriers, and the active species are firmly bonded to the substrate surface. Majjd Naderi et al. [50,51] designed a molecular sieve based on MCM–41 that comprises Cu(II) centers and amine-group-containing siloxanes chemically bound to the surface of MCM–41. A fixed-bed penetration test was carried out to measure the protection time, a parameter that was used to characterize the properties of the material. A comparison was made between the protection time of the modified MCM–41 molecular sieve and that of ASZM–T carbon against cyanide gas. As demonstrated by the data listed in Table 3, the protection time of modified MCM–41 molecular sieves against cyanide gas is longer than that of impregnated carbon.

As a result of their unique pore structures, molecular sieves display high selectivity for gas separation or elimination. However, the gas diffusion rate within said species is low. In the presence of water vapor, the performance of molecular sieves is considerable reduced due to the strong competitive adsorption of water vapor; therefore, the practical application of molecular sieves is subject to certain limitations. Especially in actual use, when a molecular sieve is employed as the filler in a fixed-bed adsorption reactor, due to the general disadvantage of the poor heat transfer in fixed–bed reactors, when the reaction heat is high, the reaction temperature may increase sharply beyond the allowable range, resulting decreased adsorption performance of the sieve.

#### 3.2.4. Magnetic Nanoparticles

In recent years, researchers have focused their attention on the study of adsorbents with tunable surfaces with the aim of using them for the adsorption of specific gases to achieve, for example, the elimination of small-molecule toxic or carbon dioxide storage [52,53,54,55,56,57,58]. Magnetic nanoparticles have received a considerable amount of attention. Not only do these nanoparticles have stable crystal structures and tunable surfaces, but their magnetic properties vary depending on the particle surface species [59].

T. Grant Glover et al. synthesized ferrite magnetic nanoparticles, such as MnFe_2_O_4_, NiFe_2_O_4_, and CoFe_2_O_4_, and used them as catalysts to eliminate CNCl in dry air [60]. The results of fixed-bed penetration experiments indicated that the adsorption of CNCl molecules by ferrite magnetic nanoparticles does not consist of simple physical adsorption. FT–IR spectroscopy data indicated that CNCl molecules and the groups present on the surface of the ferrite magnetic nanoparticles reacted chemically with each other to form carbamates. The characteristics of said reaction depend on the identity of the particulate metal components, the presence of surface hydroxyl groups, and the presence of surface-adsorbed water molecules. As is shown in Figure 4 [61]. The surface reaction product is chemically bound to the nanoparticle surface, and the nanoparticles’ magnetism is reduced as a result. According to this property, the life of the nanoparticles can be indicated by the change in material magnetism. As the nanoparticles achieve elimination of toxic and harmful gases, the consumption of their bed can be indicated by magnetic changes. However, the adsorption capacity of the nanoparticles is not high, and they lose their reactivity quickly after the surface reaction, so their applications are limited.

#### 3.2.5. Microfibrous Entrapped Composite Material

Microfibrous entrapped composite material consists of a three-dimensional network structure formed as a result of the crosslinking of microfibers; this network contains small particles with specific functions, such as particles with catalytic properties and particles affording adsorption properties. Microfibrous materials are generally selected according to actual needs—usually polymers, ceramics, and metals. The application of fiber materials to fixed beds can enhance the level of heat and mass transfer, reduce the wear of granular packing materials and the magnitude of the bed pressure drop, and increase adsorption and catalysis efficiency; therefore, these materials have broad application prospects [62].

The application of sintered microfiber-coated activated carbon in gas masks and collective protective filters has been investigated at Auburn University in the USA [63]. In particular, the penetration behavior of cyclohexane on microfiber-coated activated carbon was systematically studied. Evidence indicated that the use of sintered microfiber-coated activated carbon alleviated the problems associated with uneven fluid distribution in the granular carbon bed and slow external diffusion into micropores. Pingwei Ye et al. coated activated carbon with nickel fiber and subsequently implemented the preparation process of ASZM–T carbon to impregnate the precursor solution of copper, silver, molybdenum, zinc, and TEDA on the activated carbon covered by nickel fiber. Next, the sintered microfiber-coated, carbon-supported catalyst and ASZM–T carbon were packed into a composite bed, and the protection time of the composite bed against toxic air containing HCN was explored. As demonstrated by the data presented in Figure 5 [64], the value of the protection time of the 2.5 cm thick ASZM–T granular carbon bed is comparable to that of the 2.0 cm thick ASZM–T granular carbon bed combined with a 0.4 cm thick impregnated carbon microfiber hybrid (sintered/impregnated microfiber hybrid; SFH). SFH alone does not achieve satisfactory performance. Although a microfibrous entrapped composite material can improve the efficiency of gas diffusion and mass transfer, the presence of such a material is also associated with problems, such as powder shedding, which is induced by the weak interaction force between microfibers and activated carbon, so the applications of microfibrous cladding materials remain limited.

#### 3.2.6. Metal–Organic Frameworks

Metal–organic frameworks (MOFs) are porous crystalline materials formed by linking metal ions or metal oxide clusters with organic ligands. These materials are characterized by an ultra-high specific surface area, abundant pores, a tunable surface functional group, and a highly ordered structure [37,65,66,67]. They have a wide range of applications in many fields, such as gas adsorption [37,66] and capture [68], catalysis [69,70], gas storage [71], batteries [72], sensing [73], and sustained drug release [74].

Some MOFs are rich in unsaturated metal sites, which are active centers for adsorption and catalysis. They also contain a large number of organic ligands, which are easily subjected to functional modification. Studies have indicated that the valence state of the metal ion and the pKa value of the coordination atom of the organic ligand determines the strength of the ligand–metal coordination bond. MOFs comprising low-valence metal centers exhibit poor thermodynamic stability and are prone to structural collapse in the presence of water vapor. Zirconium-based MOFs display high thermal and chemical stabilities, and zirconium-based MOFs have good degradation effects on chemical warfare agents and their simulants, as well as on toxic industrial chemicals. As is shown in Figure 6 [37], Ga-Young Cha et al. conducted a gas-phase acid–base reaction between sublimated TEDA and the bridging hydroxyl groups on MOF–808. The TEDA-modified MOF–808 (MOF–808–TEDA) exhibited a good degradation effect on CNCl under humid conditions. Figure 5 shows the synthetic route and combination mode of MOF–808–TEDA. The results of in situ characterization analysis and density functional theory calculations revealed the formation mechanism of TEDA deposition on the zirconium-based MOFs, providing a design principle for the modification and optimization of MOFs.

MOFs are in powder form, and the direct filling of the bed causes excessive resistance when the airflow passes through. MOF powder is usually formed by the tableting method, crushed into a given number of mesh particles, and filled into a fixed bed. Peterson et al. utilized UiO–66 modified by ligand amination (to produce an MOF denoted as UiO–66–NH_2_) as the research object [35] and explored the effect of various tableting pressures on the properties of UiO–66–NH_2_. As demonstrated by the data listed in Table 4, as the tableting pressure increased, the amount of CNCl adsorbed onto the as-prepared UiO–66–NH_2_ decreased, indicating that the applied molding method of tableting influences the performance of MOFs. 

Evidence indicates that when MOF is tableted, the higher the tableting pressure, the lower the static adsorption capacity. Tableting possibly leads to the collapse of the structure, which affects the pore structure and adsorption performance of MOFs, also indicating that the material provides significant capacity but lacks the mass-transfer kinetics necessary for use in a filter. This stems from a lack of a hierarchical pore structure; further studies must be conducted to enhance diffusion characteristics of this material [35].

As demonstrated by the data listed in Table 5, the results of the dynamic penetration tests indicate that the protection time of the tableted material against CNCl is 0, and the performance is much lower than that of activated carbon [75]. Therefore, tableting is not a suitable method for engineering applications of MOFs. MOFs are suitable for engineering use in other forms, which should be further explored in future research.

Table 6 provides general information about the advantages and disadvantages of various types of materials with respect to their use for cyanide gas elimination [63,76,77]. In summary, MOFs exhibit excellent adsorption and catalytic performance. However, most MOFs are directly synthesized by various methods exist in the form of powders, requiring tableted before testing, which leads to blocked mass transfer and reduced performance. Moreover, MOFs are characterized by poor stability in water vapor, and they easily undergo decomposition in humid environments, resulting in low long-term stability [78,79,80]. Moreover, the production cost of MOFs is high, and the practical applications of these materials are limited. In recent years, MOFs/Zr(OH)_4_ composites have been prepared by the sol–gel method (Figure 7), and they have been observed to exhibit excellent properties [81]. However, the obtained species were observed to be characterized by insufficient hardness and strength; additionally, the problems of transportation and abrasion after filling must be considered, and the technology is not yet mature, so MOFs/Zr(OH)_4_ have not yet been applied for the elimination of gaseous pollutants.

## 4. Conclusions

In this paper, we reviewed various methods for cyanide gas elimination, as well as the available adsorption and catalytic materials. We discussed research progress in the development of materials to be employed in cyanide gas elimination is discussed, including carbon-based and non-carbon-based materials. Moreover, the advantages and disadvantages of various materials were summarized. Based on the existing data, suggestions for the direction of future research work are as follows:(1)The adsorption of cyanide gas onto porous materials is weak, and the mechanism of the adsorption and elimination process is unclear. The cyanide gas elimination performance of existing materials is not satisfactory. A new material system needs to be designed in the future based on a deeper understanding of the mechanism of the adsorption and elimination process. First, a hierarchically porous material matrix with a mesoporous and microporous composite structure should be constructed to reduce the mass transfer resistance of gas on the matrix material. Secondly, a large number of active sites should be constructed on the matrix material. After the cyanogen gas is adsorbed, it can react and degrade rapidly.(2)Compared with other materials, MOFs exhibit structural diversity and designability, and they are characterized by broad application prospects for the efficient adsorption and in situ degradation of chemical warfare agents. Future research should focus on establishing a method for the practical application of MOFs that reduces their performance degradation.(3)Porous organic framework (POF) materials have a large surface area, functionable channels, and tunable pore size and are highly environmentally stability. Such materials contain a high density Lewis acid boron sites that can strongly interact with Lewis basic guests, making them ideal for the storage of corrosive chemicals, such as ammonia [82]. CJ Doonan et al. [83] found that a member of the covalent organic framework family, COF-10, a higher uptake capacity (15 mol·kg^−1^, 298 K, 1 bar) than any porous material, including microporous 13X zeolite (9 mol·kg^−^^1^), Amberlyst-15 (11 mol·kg^−^^1^), and mesoporous silica MCM-41 (7.9 mol·kg^−^^1^). Furthermore, POFs exhibit high removal efficiency for nerve agents [84]. Therefore, we speculate that POFs have the potential to be used for the removal of cyanide gas.(4)Catalysis is a suitable method for the elimination cyanogen chloride. The following principles can be applied in the design of high-efficiency catalysts. First, the catalyst is highly active. Secondly, the stability is sufficient to bear high concentrations of oxygen, carbon dioxide, and water in the case of practical application. Thirdly, catalyst poisoning can be avoided or reduced to ensure a long catalyst service life. Finally, the preparation of the catalysts should be easy low-cost.(5)On the basis of maintaining the stability of the catalyst, improving the activity of the catalyst should be the focus of future work. If the intrinsic activity of the catalyst can be considerably improved, the temperature required for the catalytic process can be reduced, which is beneficial in terms of energy conservation. In addition, if the intermediate products or products can be chemically adsorbed on the active parts of the catalyst surface to form strong adsorption species, catalyst poisoning may result. Therefore, effective inhibition of catalyst poisoning should also be a focus of future research to ensure long catalyst life.

## Figures and Tables

**Figure 1 molecules-27-07125-f001:**
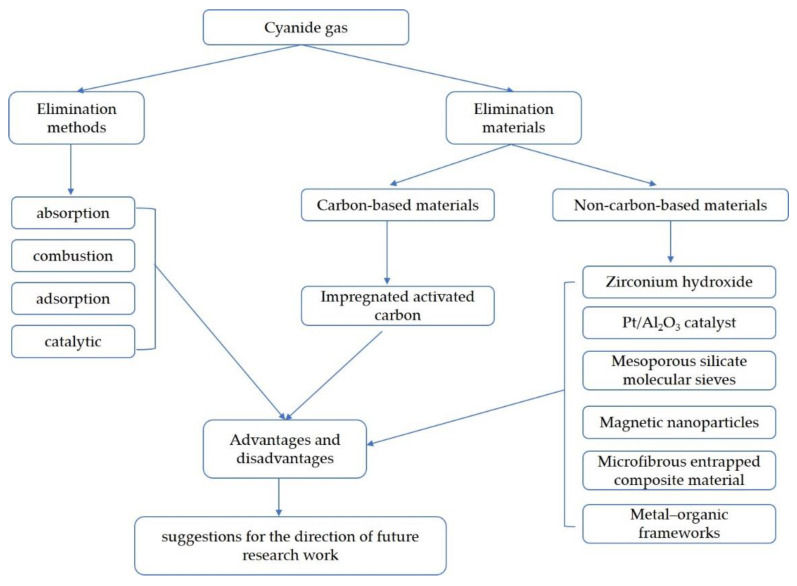
The scope of this review.

**Figure 2 molecules-27-07125-f002:**
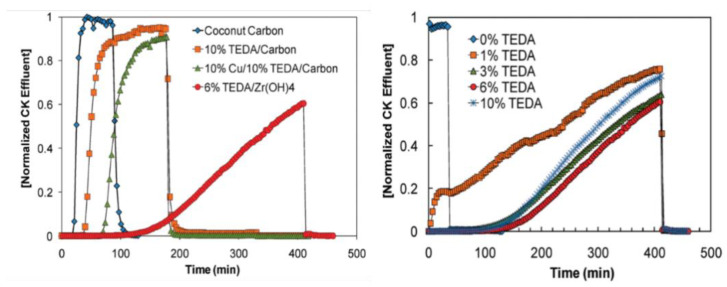
Penetration curves of CNCl on different protective materials [36].

**Figure 3 molecules-27-07125-f003:**
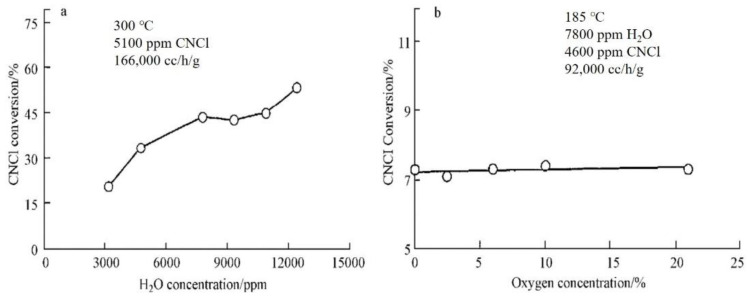
Effects of water (**a**) and oxygen (**b**) concentrations on the Pt/Al_2_O_3_-catalyzed thermal decomposition of CNCl [49].

**Figure 4 molecules-27-07125-f004:**
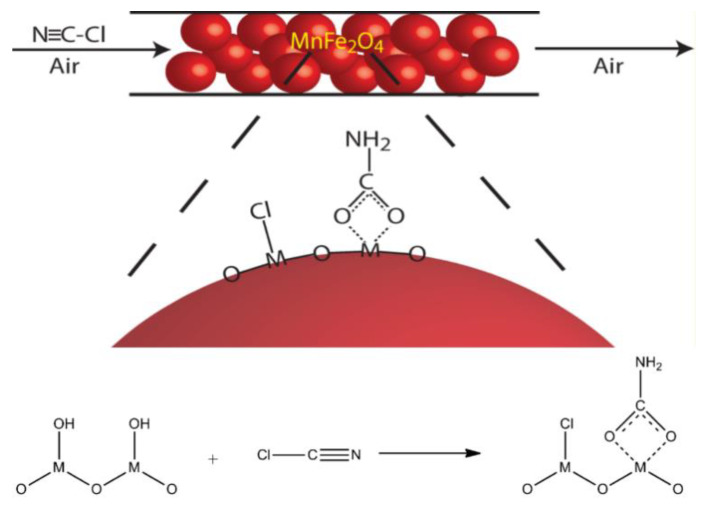
Mechanism of the surface reaction between CNCl and magnetic nanoparticles [61].

**Figure 5 molecules-27-07125-f005:**
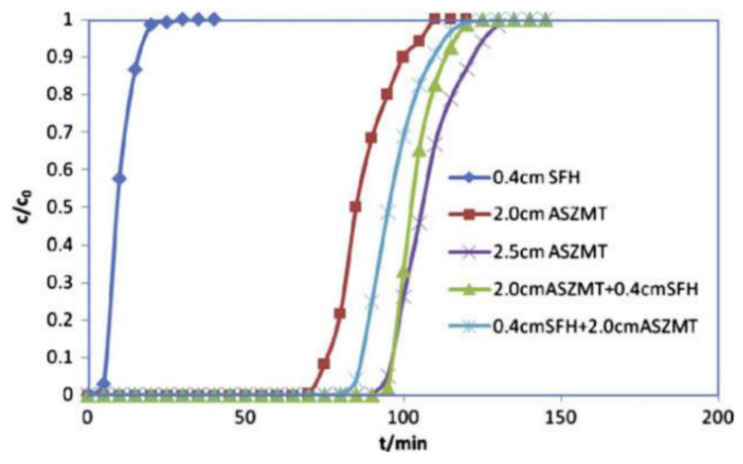
HCN penetration curves determined for microfiber-coated composites prepared by implementing different filling methods and ASZM–T granular carbon composite fixed beds. The initial concentration of HCN was 4 mg·L^−1^, and the gas flow rate was 0.79 L·min^−1^ [64].

**Figure 6 molecules-27-07125-f006:**
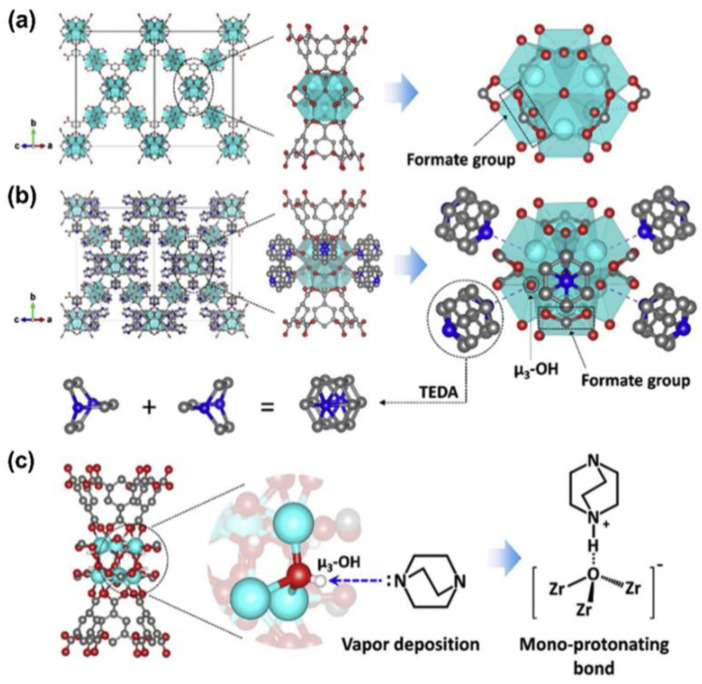
Synthetic route and combination mode of MOF–808–TEDA. (**a**) MOF–808; (**b**) T–MOF–808. The polyhedrons represent ZrO_8_ groups. Atom colors: Zr: cyan; C: dark gray; O: red; and N: blue. Water molecules were excluded. (**c**) Schematic representation of TEDA (triethylenediamine) functionalization of MOF–808 [37].

**Figure 7 molecules-27-07125-f007:**
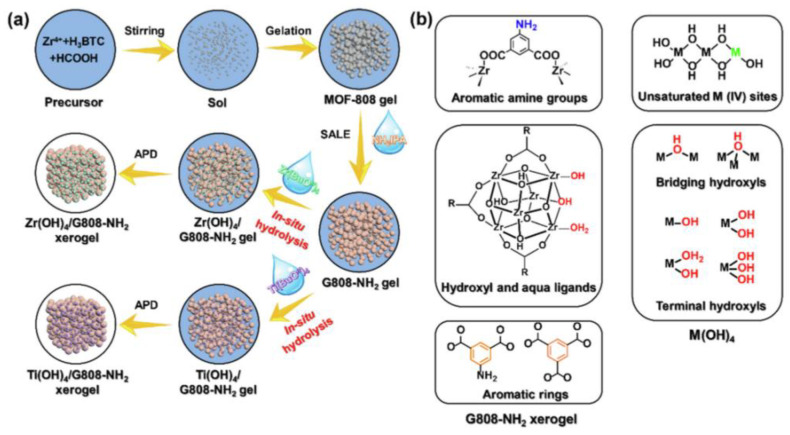
(**a**) Scheme for the synthesis of M(OH)4/G808–NH_2_ (M = Zr, Ti) xerogel composites via an in situ hydrolysis strategy; (**b**) potential sites for irreversible adsorption for poisonous species [81].

**Table 1 molecules-27-07125-t001:** CNCl capacities of different materials.

Material	Breakthrough Time (min)	Capacity (mg/cm^3^)	Test Parameters	Ref.
Carbon	14.8 ^*a^	/^*c^	Particle size: 12 × 30 mesh Bed depth: 2.0 cm Volume: 26.4 cm^3^ Air flow velocity: 6.0 cm/sAir flow rate: 4.7 L/min Temperature: 24 ± 2 ℃ Relative humidity: 80% Challenge concentration: 4000 mg/m^3^ Breakthrough concentration: 5 mg/m^3^	[35]
10%TEDA/carbon	33 ^*b^	39		[36]
10%Cu/10%TEDA/carbon	61 ^*b^	67		
6%TEDA/Zr(OH)_4_	87 ^*b^	231		

*a: CNCl breakthrough time was calculated at the breakthrough point ([CK] = 8 mg/m^3^). *b: CNCl breakthrough time and capacities were calculated at the breakthrough point ([CK] = 5 mg/m^3^) *c: The CNCl capacity of carbon has not been reported in public literature. This research work will be conducted by our group in the future.

**Table 2 molecules-27-07125-t002:** CNCl capacities of MOFs [37].

Materials *	Adsorbed Amount of CNCl * (mmol/g)	
	Dry	Humid (293K, RH80%)
T–MOF–808	1.26	4.05
T–UiO–66	1.43	2.38
T–MIL–100	0.31	0
T–MIL–101	1.78	0
T–ZrO(OH)2	1.19	1.28
T–ASZM	1.07	1.19

* CNCl capacities were calculated at the breakthrough point ([CK] = 5 mg/m^3^); * T–Materials means materials modified with TEDA.

**Table 3 molecules-27-07125-t003:** Penetration time of HCN, CNCl, and (CN)_2_ on MCM–41 and carbon-based adsorbent [51].

Adsorbent	Cu^2+^/%	HCN/min	CNCl/min	(CN)_2_/min
SiMCM–41–en–Cu^2+^	2.4	28	64	26
Carbon–Cu–Cr–TEDA	7.0	24	29	22

**Table 4 molecules-27-07125-t004:** Protective performance of UiO–66–NH_2_ formed with varying tableting pressure on CNCl [35].

Sample	RH (%)	Mass (mg)	Adsorption Capacity of CNCl (mol/g)
UiO–66–NH_2_	0	7.9	4.1
	80	7.8	1.2
UiO–66–NH_2_–5K	0	10.8	3.9
	80	15.5	1.0
UiO–66–NH_2_–10K	0	11.3	2.9
	80	11.8	0.5
UiO–66–NH_2_–25K	0	18.3	2.7
	80	20.5	0.9
UiO–66–NH_2_–100K	0	33.2	0.8
	80	40.0	0.3

Note: UiO–66–NH_2_–5K defines a sample obtained by subjecting UiO–66–NH_2_ powder to a pressure of 5000 psi, and the meanings of other MOF names are analogous.

**Table 5 molecules-27-07125-t005:** Comparison of breakthrough time and protective dose of CNCl between UiO–66–NH_2_–5K and impregnated carbon-based carbon [35].

Chemical	Adsorbent	Challenge Concentration (mg/m^3^)	Penetration Concentration (mg/m^3^)	Breakthrough Time (min)
CNCl	Activated carbon	4000	8	14.8
	UiO–66–NH_2_–5K			0

**Table 6 molecules-27-07125-t006:** Advantages and disadvantages of various materials for use in cyanide gas elimination.

Type of Material	Advantages	Limitations
Impregnated carbon	Rich in micropores Strong physical adsorption performance. Good protection against macromolecular poisons.	Poor protection against cyanide-based small-molecule poisons. Disordered pores. Structure is difficult to design and control.
Zr(OH)_4_	Surface is acid–base amphoteric. Easily modified according to need.	Powdery. Difficult to shape.
Pt/Al_2_O_3_ catalyst	Excellent broad-spectrum elimination performance. High efficiency.	Heating necessary. High energy consumption.
Mesoporous molecular sieves	Regular pore structure and long-range order.	Pore structure is fixed. Difficult to regulate according to demand.
Microfibrous entrapped composite material	Enhanced heat and mass transfer. The magnitude of the bed pressure is reduced. Increased adsorption and catalysis efficiency.	Very limited adsorption capacity toward small molecules.
Magnetic nanoparticles	Magnetic changes indicate bed consumption	Low adsorption capacity.
Metal–organic frameworks	High specific surface area and abundant pores. Easily modified surface. High structural order. Adjustable channels.	Poor adsorption performance when piled up. Poor water vapor stability.

## Data Availability

Not applicable.

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
