# Peer review of "A Review on Cyanide Gas Elimination Methods and Materials"

_molecules, 2022, doi:10.3390/molecules27207125_

Round 1

Reviewer 1 Report (New Reviewer)

In this paper, the authors review the currently implemented methods for cyanide gas elimination, discuss the development of materials towards cyanide gas elimination, and outlook the further research direction in this field. The paper is well organized, exhaustive and comprehensive, and may serve as a competent guide for researchers to rapidly learn what has been done and what is yet to be done in this field. As cyanide gas elimination is an emerging and important scientific topic, I am supportive of its publication. Here are some suggestions for improving this review.

The authors summarize the different materials used for cyanide gas elimination and explain their advantages and disadvantages. Could the authors elaborate further on what different specific cases these materials are used in?

It seems that there are few catalysts can be used for cyanide gas elimination. Could the authors suggest guidelines for designing efficient catalysts? If the intrinsic activity of the catalyst can be greatly improved, does it mean that the temperature required for the catalytic process can be reduced? If so, in that case the elimination of cyanide gas by catalysis looks very competitive.

Could the author further enrich the outlook of catalytic cyanide gas elimination?

Author Response

Reviewer 2 Report (New Reviewer)

This paper reviewed various cyanide gas elimination methods and the different adsorption and catalytic materials. In terms of carbon-based and non carbon-based materials, the research progress in developing materials to be employed in cyanide gas elimination is discussed; moreover, the advantages and disadvantages of various materials are summarized.

Although the manuscript is well presented, before accepting the paper for publication I suggested to the authors to introduce some minor changes to improve the paper. The authors need to provide or add a wider description of gas elimination methods which can be viewed from several aspects such as economic, environmental, practicality, and several other important aspects.

Author Response

This manuscript is a resubmission of an earlier submission. The following is a list of the peer review reports and author responses from that submission.

Round 1

Reviewer 1 Report

A review on cyanide gas elimination methods and materials

Manuscript ID: molecules-1874270

This manuscript introduced a review of method and materials for cyanide gas elimination. The authors provide comprehensive and useful knowledges about the uses of different adsorbents in cyanide gas capture as well as highlights the advantages and disadvantages of each type of materials. However, several parts in the manuscript are lacking supporting numerical data and proper citations. The outlook of this manuscript is also not well-discussed in depth. Hence, a major revision is recommended for this manuscript with some important comments listed below:

Section 1: Introduction

1. Line 18: the full names of HCN, CNCl and (CN)2 should be detailed in the first time of use and the chemical formulas will be used in the rest of manuscript (line 26, 142, 1489, 301, 305). In the current manuscript, the use of chemical names and formulas are mixing and random.

2. Line 21-24: Please add the references for each source of cyanide gases.

(Please consider that the paper is for general readers who might not be an expert in the field. Any information obtained from the literature or not well-known should have a citation right next to it. The first paragraph of section 3.2.6 has a good citation, please use it as an example to revise other parts of the manuscript)

3. Line 31: Please add reference

4. Line 34: How are the used adsorbents eliminated/disposed/decomposed currently?

5. Line 36: what do authors mean ‘the adsorption of cyanide gas is difficult to realize…’?

6. Line 37: Please add a reference about the impact of cyanide gas in environment

7. Are there any literature about cyanide gas elimination reported in the literature?

Section 2: Cyanide gas elimination methods

8. Line 55: …between the various components ‘with the absorbent’?

9. Line 67-69: ‘Indeed, the relevant processes … of the equipment” – this sentence is unclear, please rephrase

10. Line 70-72: Is this sentence a view of authors about absorption method or a summary from reference [2]?

11. Can authors make a summary table about the sources of cyanide gas and the potential gases (besides cyanide gas) in each source? This will affect which technique can be used for cyanide gas treatment. For instance, if the elimination method relies on alkali absorbents/adsorbents, some acidic gases such as CO2, HCl will affect the performance of HCN elimination.

12. Line 76-77: “By this approach, not only can harmful gases be prevented from polluting the environment…” – this writing structure is not readthrough, suggest rephrasing

13. Line 86: Please provide numerical temperature range for direct combustion method

14. Line 86-87: what do authors mean “it affords no selectivity for 86 gas conversion”?

15. Line 95: ‘…the relevant catalysts lack environmental stability’ – do authors mean the low-cost catalysts will be degraded in environment? What environment do authors refer to?

16. Line 99: Similar to Comment 10. If this summary sentence is from ref [4], what is authors’ critical view about combustion method?

17. Line 112-117: This sentence is combining 3 sentences, please split it out. (Using semicolon to connect more than two sentences makes then sentence too long and difficult to readthrough)

18. Line 117 – 119: Can authors detail what the goal of harmless gas is? Are we targeting toward the legal airborne permissible exposure limit or any other targets?

Section 3: Cyanide gas elimination materials:

19. Figure 1, 2, 5: It is unclear what academic information the authors wish to inform in these figures. Recommend combining them into one figure. Also, are these figures originally produced by the authors or from other sources? If the latter, a proper citation must be included in the figure name. The authors will need the Copyright Permissions from the original sources of the figures to reproduce them in this paper as well.

20. Line 147 – 150: Similar to comment 10 & 16

21. Line 154: Do we have a full name of ASC? If yes, please write it in full at the first use

22. Line 151 – 172: the whole paragraph does not have any reference. Please add them in the proper locations.

22. Line 161 – 167 & Line 167 - 173: Similar to Comment 17, please prevent using semicolon to connect 3 or more sentences.

23. Line 174: Reference is required for the stop issuing licenses for chromium-impregnated carbon.

24. Can authors include a table summary the adsorption capacity for cyanide gas of different types of adsorbents? Without the support of numerical data, the comparison between different adsorbent is too skeptical and subjective.

25. Line 212-219: Similar to comment 17 & 22, please split the sentence into shorter sentences.

26. Line 212: Reference required for this statement. Also, some data of porosity and surface areas versus raw materials of activated carbon will be helpful to support this statement.

27. Line 219-222: reference required

28. Line 245 – 254: Ref 18 is about NH3 and SO2 separation and irrelated to cyanide gas elimination. Why is it included in this section?

29. Figure 3,4, 6,7, 8,9: Are these figures originally produced by the authors or from other sources? If the latter, a proper citation must be included in the figure name and the authors will need the Copyright Permissions from the original sources of the figures to reproduce them.

30. Line 277: Zirconium hydroxide is better in cyanogen chloride elimination than ‘  ‘?

31. Section 3.2.2: While other 3.2.x subsection focus on cyanide gas adsorption, section 3.2.2 is about catalytic decomposition method.

+ Should authors include some review about the adsorption performance of Pt/Al2O3 catalyst first prior discussing its performance in cyanide gas decomposition?

+ How are the used Zr(OH)4 , MOFs,… disposed? Heat treatment as well?

32. Line 299: please confirm Hads CNads or H+ads CN-ads

33. Line 306 – 308: reference required. Please also be consistent in the location of citation, either after the authors’ names or at the end of the sentence.

33. Figure 4 is likely reproduced from Agarwal, S. K., Spivey, J. J., & Tevault, D. E. (1995). Kinetics of the catalytic destruction of cyanogen chloride. Applied Catalysis B: Environmental5(4), 389-403. This reference isn’t included in the manuscript

34. Should the title of section 3.2.3 be ‘Mesoporous silicate molecular sieves’?

35.Line 351: please remove ‘(HCN, CNCl, and (CN)2)’. This repeat Line 18.

36. Line 381 – 383: ‘According … can be monitored’ – this sentence is confusing, please rephrase

37. Line 411 – 415: Please recheck this sentence. The interpretation of authors about Figure 7 is not convinced and confusing. The breakthrough time of 2.0 and 2.5cm ASZMT are around 75 and 100 minutes, which are not comparable. SFH along is very bad performance so it is also not comparable with others.

38. Table 2 – being similar to other figures, the citation is required either in the table name or next to the corresponding data within the table.

39. Are there any other MOFs used for cyanide gas separation? If so, should their adsorption capacity be included into Table 2 as a comparison between MOFs?

40. Line 473 – 475: Should authors conclude that MOFs are not suitable for protecting engineering use?

Conclusions:

41. The outlook of authors about the future research are too short and too general for a review paper. Please expand authors’ view in more details

42. POFs were not mentioned in the previous sections and it is unclear how good POFs compared to other adsorbents. Can authors provide some numerical data to address this? (the data for cyanide is ideal but for other gases are also acceptable)

43. Table 4 is a good comparison. However, the whole review doesn’t have much numerical results about the performance of cyanide gas sorption, especially for carbon-based adsorbents. As so, Table 4 alone can’t provide practical guidance in adsorbent selection.

44. Reference:

+ Please check the reference style to match journal’s requirement (e.g., ref 3, 21)

+ Only 23% (15 out of 63) references are published after 2015. Should authors update some recent publications into this paper?

Reviewer 2 Report

Review report for molecules-1874270

The subject is attractive enough, however, the following points should be addressed before further considerations.

1-    What was the authors motivation of writing this manuscript?

2-    Were there any review papers on this subject? If so, did the authors include in the manuscript?

3-    Please insert the related references for all figures and tables used in the manuscript.

4-    As stated one of the cyanide gas elimination methods is absorption.  Authors are strongly recommended to use the following reference (10.1016/j.icheatmasstransfer.2021.105193).

5-    English language should be double-checked.

Reviewer 3 Report

Shuyuan Zhou and Shupei Bai presented a review paper on cyanide elimination techniques for consideration. This gas is very toxic, and some crucial manufactures are accompanied by its release. The elimination or neutralization of this gas is an important task that can be solved using various approaches. The current approaches are considered in the review.

Unfortunately, I cannot recommend this work for publication in Molecules for the following reasons:

1. Any review should give a clear picture of the existing problem and current methods of solving it, with an appropriate critical analysis. In this review, I found some not clarified data, inaccurate wording, for example, in the conclusions, where the authors discuss current problems in a very abstract way.

2. Some of the data given are obvious (eg cyanide adsorption). I see no point in discussing this, since this is not about a textbook.

3. Many of the facts presented are not supported by references. For example, there is no refs in the Introduction. If these are simple truths that do not require proof, then it should not be cited, because it is common knowledge. If this is so much specific information, then it should not be given either, because a reader will not be able to verify this data anywhere.

4. Many key points are missing. For example, there is no clear explanation of what is new in this review and what has already been done before the authors. Typically, the first scheme (or first figure) shows progress in a key area, explaining the contribution of the authors. In the current version, it is difficult to understand what is new and what has been done for early.

5. Some figures are too common and useless. For example, a figure showing activated carbon. Is this really the information that is unknown? Will it be of interest to a reader? The first figure is the main figure, so, it’s better to show something important. And there are many such figures: fig 2 (zirconium hydroxide, it’s just a white powder); fig 5 (just a white spheres).

6. The review would be better if the English language was better checked. There are a number of suggestions that could be improved; e.g., In this paper are reviewed the cyanide gas elimination methods that are currently implemented should be reconsidered according to the direct words order.

7. Unfortunately, I managed to find a lot of inaccuracies and tips. For example, the list of references is compiled with violations, which does not allow finding the necessary information; e.g., Gregory; Mogilevsky; and; Christopher; J.; Karwacki; and; Gregory; W.; Peterson, Surface hydroxyl concentration on Zr(OH)4 quantified by 1H MAS NMR. Chemical Physics Letters 2011

62. Xw, A.; Li, L. A.; Kai, L. A.; Rs, A.; Yue, Z. A.; Song, G. B.; Wg, B.; Zl, A.; Gl, A.; Hx, A., Hierarchically porous metal hydroxide/metal–organic framework composite nanoarchitectures as broad-spectrum adsorbents for toxic chemical filtration. Journal of Colloid and Interface Science 2021 (where is the pages, vol., numbers?)

Round 2

Reviewer 1 Report

The manuscript has been well-addressed comments from reviewers and my recommendation is to accept the manuscript in the current form.

P/S: There is one small comment for Line 434 which is unclear what "life" the authors are referring to (life of nanoparticles?). Good to address it clearer.

Reviewer 2 Report

Authors have responded and addressed the reviewer comments. Accordingly, I am now convinced to accept the manuscript.

Reviewer 3 Report

I found some changes in the manuscript. However, the changes are minor and insufficient.

-        Tables 1 and 2 should be summarized information (as the authors answered). In fact, the summary in table 1 is only 2 refs and the summary in table 2 is only one ref. It is not summary, and it is not necessary to transform the text from one paper to table.

-          The ref list still involves mistakes: 10, 19, 25-26 (not available info). The names of journals should be abbreviated.

-          The text marked in yellow – is not new. The new text is in red (not many changes, just minor corrections)

-          Figures 1, 2 etc should be replaced, but not simply removed. The progress, limitations, novelty, main ideas or any crucial info should be added to the text (like TOC) on the first pages to introduce the subject of the review.

-          “a prospect for our future work” looks inappropriate, since it is a review. The better choice is to predict the future direction of the field for the chemical society (but not for the authors). It looks fine in the case of standard research paper.